# Polyhydroxyalkanoate Production by *Caenibius tardaugens* from Steroidal Endocrine Disruptors

**DOI:** 10.3390/microorganisms10040706

**Published:** 2022-03-24

**Authors:** Juan Ibero, Virginia Rivero-Buceta, José Luis García, Beatriz Galán

**Affiliations:** Centro de Investigaciones Biológicas Margarita Salas, 28040 Madrid, Spain; jicaballero@cib.csic.es (J.I.); mvrivero@cib.csic.es (V.R.-B.); jlgarcia@cib.csic.es (J.L.G.)

**Keywords:** steroids, estrogen, androgen, *C. tardaugens*, polyhydroxyalkanoates, polyhydroxyvalerate

## Abstract

The α-proteobacterium *Caenibius tardaugens* can use estrogens and androgens as the sole carbon source. These compounds are steroidal endocrine disruptors that are found contaminating soil and aquatic ecosystems. Here, we show that *C. tardaugens*, which has been considered as a valuable biocatalyst for aerobic steroidal hormone decontamination, is also able to produce polyhydroxyalkanoates (PHA), biodegradable and biocompatible polyesters of increasing biotechnological interest as a sustainable alternative to classical oil-derived polymers. Steroid catabolism yields a significant amount of propionyl-CoA that is metabolically directed towards PHA production through condensation into 3-ketovaleryl-CoA, rendering a PHA rich in 3-hydroxyvalerate. To the best of our knowledge, this is the first report where PHAs are produced from steroids as carbon sources.

## 1. Introduction

Natural estrogens and androgens enter the environment through the excretions of humans, domestic or farm animals, and wildlife. 17β-estradiol (E2) (estrogen) and testosterone (TES) (androgen) are the most ubiquitous sexual hormones found as pollutants in soil and water systems [1,2,3,4]. The release of hormones in the environment from sewage treatment plants, livestock feedlots, or hospital effluents causes adverse effects on aquatic life, e.g., by affecting physiology, behavior, and sexual development of fish [5]. In particular, estrogens are classified as group 1 carcinogens by the World Health Organization (https://monographs.iarc.fr/list-of-classifications/, accessed on 10 February 2022).

Microbial biodegradation is considered the major process for the elimination of estrogens and androgens from aquatic environments. Estrogens are more recalcitrant than androgens, but some specialized microorganisms have adapted to use them as sole carbon and energy sources (mineralization) or, at least, partially degrade these molecules to less-toxic and persistent compounds [6]. *Caenibius tardaugens* was isolated in a sewage treatment plant in Tokyo (Japan) due to its ability to use steroid endocrine disruptors such as estrone, E2 and estriol as sole carbon and energy sources [7]. Recently, we have elucidated the catabolic pathways of *C. tardaugens* involved in the degradation of TES and E2 [6,8,9,10]. The aerobic degradation of TES and E2 is initiated by the dehydrogenation of the 17β-hydroxyl group to androst-4-en-3,17-dione (AD) and to estrone, respectively. Then, AD degradation proceeds via the 9,10-*seco* pathway, while estrone is further catabolized using the so-called 4,5-*seco* pathway [6,11,12,13,14].

Polyhydroxyalkanoates (PHAs) are biodegradable and biocompatible linear polyesters of (R)-3-hydroxycarboxylic acids, which are naturally produced by some microorganisms, as cytoplasmic inclusions, under conditions of nutritional imbalance [15]. These materials have been actively studied as promising materials for solving issues related to the currently used plastics because they possess physical and thermochemical material properties comparable to those of petroleum-based plastics [16]. The material properties of PHAs can vary depending on the monomer chain length and composition. In this sense, PHAs can be divided into two general categories, namely short-chain length (SCL)-PHAs and medium-chain length (MCL)-PHAs. SCL-PHAs are composed of monomers comprising three to five carbon atoms, representative examples of which are poly(3-hydroxybutyrate) (PHB) poly(3-hydroxyvalerate) (PHV) and poly(3-hydroxybutyrate-co-3-hydroxyvalerate) (PHBV). There is increasing interest in producing these polymers using different wastes as substrates mainly with the aim of reducing PHA production costs [16,17]. These feedstocks include lignocellulosic residues, sugar cane molasses, palm oil, waste frying oils, crude glycerol, waste office paper hydrolysate, etc. [18]. In addition, with recent advances in the biorefining of C1 gasses such as carbon monoxide (CO), carbon dioxide (CO_2_) and methane (CH_4_), PHA production from these gasses has been actively studied [19].

The bio-based production from contaminants could be turned into efficient and economic bioprocesses, combined with the respective social and environmental benefits. In this sense, PHAs can be produced from different carbon sources considered as contaminants; thus, the same compound that is targeted for elimination could be used for the synthesis of the polymer contributing to the persistence of the bioremediation agent in the environment and, consequently, the overall efficiency of the process. There are only a few examples in which PHA production has been connected to decontamination strategies, including different hydrocarbons such as benzene, toluene, and xylene (named as BTX), styrene and phthalates [16,20,21,22,23,24]. PHA accumulation using activated sludge treated wastewater also has been explored, resulting in PHA production of up to 29% of dry cell weight [25].

To our knowledge, endocrine disruptor steroidal contaminants have never been tested as substrates for PHA production. Only a few species within the estrogen degraders family Sphingomonadaceae are able to accumulate PHAs, including two bacteria of the genus *Novosphingobium* [26,27,28], a genus in which *C. tardaugens* was previously included [29].

In this work, we analyzed the ability of *C. tardaugens* to grow on different environmentally relevant steroids and evaluated its application as a biocatalyst for steroid valorization towards the synthesis of (SCL)-PHAs.

## 2. Materials and Methods

### 2.1. Chemicals

Testosterone (TES) and 17β-estradiol (E2), methanol, sulfuric acid, chloroform and methyl benzoate were purchased from Sigma (Steinheim, Germany). Randomly methylated β-cyclodextrin (TRMB-T Randomly Methylated BCD) (CDX) was purchased from Cyclodex (Alachua, FL, USA). Poly(3-hydroxybutyric acid-co-3-hydroxyvaleric acid) (PHBV) and other chemicals and reagents were purchased from Merck KGaA Sigma (Darmstadt, Germany).

### 2.2. Strains and Culture Conditions

*C. tardaugens* NBRC 16725 (strain ARI-1) was purchased from the Leibniz-Institut DSMZ type culture collection. Nutrient broth (NB) (DifcoTM, Burlington, NJ, USA) was used as rich medium to grow this strain at 30 °C in flasks using an orbital shaker at 200 rpm. PHA production experiments to determine PHA content were performed in 250 mL flasks, while those for NMR analysis where performed in 1 L flasks. This strain was also cultured in minimal medium M63 [KH_2_PO_4_ (136 g L^−1^), (NH_4_)_2_SO_4_ (20 g L^−1^), FeSO_4_ 7H_2_O (5 mg L^−1^), pH 7.0] supplemented with 0.39 mM CaCl_2_, 1 mM MgSO_4_ and different carbon sources. Steroids were dissolved in methyl-β-cyclodextrin 70 mM (CDX) (TRMB-T Randomly Methylated-Beta-Cyclodextrin, CTD Inc.). Steroid stock solutions were prepared in PBS buffer and 70 mM CDX at the following concentrations: 10.5 mM E2 and 10 mM TES. Thus, in the cultures, the final concentration was 13.33 mM CDX and equimolar amounts of carbon sources in all cases.

To produce PHA, *C. tardaugens* was grown on M63 0.1 N, which is a nitrogen-limited variation of M63 medium containing 0.2 g L^−1^ of (NH_4_)_2_SO_4_ [30], as well as on NB-rich medium.

### 2.3. Polymer Analysis

To determine the total PHA content, biomass from 65 mL of culture was collected, lyophilized and subjected to a methanolysis reaction to be subsequently analyzed by GC coupled to mass spectroscopy, following the method described previously [31]. First, 2 to 5 mg of lyophilized culture, 2 mL of methanol with 3% H_2_SO_4_ and 2 mL of chloroform with 0.5 g L^−1^ of methyl benzoate (internal standard) were introduced into a 12 mL Pyrex tube. The tube was hermetically sealed with Teflon cap and immersed in an oil bath at 100 °C for 4 h with magnetic stirring for the acid methanolysis reaction of PHA to take place. Subsequently, the samples were placed on ice, the contents were transferred to 15 mL Falcon tubes, and the stirring magnet was removed. Subsequently, 1 mL of deionized water was added, shaken for 30 s, centrifuged (15 min at 3800× *g* and 4 °C), and the upper aqueous phase was removed. The extraction of the aqueous phase was repeated twice. Finally, the organic phase was dehydrated with the addition of solid anhydrous sodium sulphate (Na_2_SO_4_). The obtained organic phase was analyzed on a Perkin Elmer AutoSystem GC gas chromatograph equipped with a flame ionization detector and a Supelco SPB1 column (20 m × 0.25 mm i.d. × 0.22 μm). The volume injected was 1 μL. In addition to the samples, the same procedure was performed with known amounts of pure PHBV, as a standard and internal control of the methanolysis process. The GC conditions were: initial temperature of 60 °C for 2 min and analysis ramp of 15 °C min^−1^ up to 220 °C in the oven and 300 °C in the detector and injector.

### 2.4. Characterization of PHA Polymers by Nuclear Magnetic Resonance (NMR)

PHA extraction was performed following the protocol described previously [32]. First, biomass from 125 mL of culture was harvested and lyophilized. Subsequently, the polymer was extracted by adding 10 mL of chloroform and heating at 90 °C for 4 h with magnetic stirring. The organic phase was evaporated under hood until 5 mL remained. The polymer was purified by precipitation with 10 vol. of cold methanol and manual stirring. The mixture was centrifuged (8500× *g*, 15 min), and the supernatant containing the organic solvents was removed. Finally, the organic solvent residues were evaporated by hood evaporation. The NMR spectrum was recorded at 500 MHz with a Bruker AV 500 MHz instrument (Sikerstrifen, Germany) at room temperature using deuterated chloroform as solvent.

### 2.5. Transmission Electron Microscopy (TEM)

Sample preparation for electron microscopy was performed on *C. tardaugens* cells grown in NB rich medium and M63 0.1 N minimal medium with E2 and TES as carbon sources until late exponential phase. Cells were washed 2 times with PBS and subsequently processed at the Electron Microscopy Service of the Centro de Investigaciones Biológicas Margarita Salas (CIBMS-CSIC). They were initially fixed for 1 h at room temperature in a 3% solution of glutaraldehyde prepared in PBS. After the fixation process, 3 washes of 10 min each were performed in PBS, and post-fixation was performed for 1 h at 4 °C in a 1% solution of osmium tetroxide and 0.8% potassium ferricyanide in PBS. After post-fixation, 3 washes of 10 min each in PBS were performed and dehydrations were started in increasing gradients of absolute ethanol: 30%, 50%, 70%, 80% and 90%. Subsequently, the inclusion in LX112 resin was performed in 72 h gradients in which the ethanol:resin ratio was increased following the ratio: 2:1, 1:1, 1:1, 1:2 and total resin twice. This was followed by encapsulation for 48 h at 60 °C to harden the resin. The samples thus prepared were processed with the ultramicrotome (Reichezt-Jung, Reichert Technologies Life Sciences, New York, NY, USA) producing 60–80 nm cuts and adhered to the grid. Copper grids of 75 MESH were used. Finally, the samples were counterstained with a 5% uranyl acetate solution for 15 min followed by a lead citrate solution for 2 min and allowed to dry. The samples were visualized using a JEOL JEM-1230 transmission electron microscope (Jeol Ltd., Tokyo, Japan) and a 16 MegaPixel CMOS digital camera, TemCam F416 (TVIPS GmbH, Gauting, Germany).

### 2.6. Bioinformatics Analysis

The amino acid sequences of individual proteins were compared with those in the databases using the BLASTp program [33] from the National Centre for Biotechnology Information server (NCBI; https://blast.ncbi.nlm.nih.gov/Blast.cgi accessed on 10 February 2022).

## 3. Results

### 3.1. Analysis of Genes Related to PHB Synthesis in C. tardaugens

Although the ability to accumulate PHAs is a widely distributed phenotype within prokaryotic organisms [34], only a few species within the Sphingomonadaceae family are reported to accumulate PHAs, including two of the genus *Novosphingobium* [26,35].

We searched for genes that could encode proteins involved in the production of PHAs in *C. tardaugens* by comparing the genome-annotated proteins with the proteins involved in the synthesis of PHAs in *Cupriavidus necator* H16 [36]. The synthesis of PHB in *C. necator* H16 involves three main enzymatic steps: the condensation of two acetyl-CoA molecules into one acetoacetyl-CoA molecule by a β-ketothiolase (PhaA), its subsequent reduction to (R)-3-hydroxybutyryl-CoA by an NADPH-dependent acetoacetyl-CoA reductase (PhaB), and PHB polymerization by ester bonding of the monomers by a synthase (PhaC) [37]. Other proteins involved in the metabolism of PHAs are phasins, which are granule-associated proteins involved in the stabilization of the PHA granules [38], and PHA depolymerases (PhaZ), which are responsible for polymer degradation [39].

First, we identified three putative PhaC enzymes encoded in *C. tardaugens* genome, i.e., PhaC1 (*EGO55_12105*), PhaC2 (*EGO55_17580*) and PhaC3 (*EGO55_18530*) (Table 1), similar to Class I PhaC enzymes that are preferentially active towards (*R*)-3-hydroxyacyl-CoAs containing three to five carbon atom acyl chains, suggesting that it might render PHBV polymers.

Moreover, there are nine putative β-ketothiolases in *C. tardaugens* genome (Table 1), suggesting that this microorganism will be able to produce different PHA copolymers, depending of the β-ketothiolase substrate specificity. For instance, formation of PHBV requires the additional condensation of acetyl-CoA with propionyl-CoA to form 3-ketovaleryl-CoA. We found two putative acetoacetyl-CoA-reductases encoded by *EGO55_18540* and *EGO55_09490* genes and one putative phasin, named PhaP, encoded by *EGO55_12100* (Table 1). Mobilization of PHA during carbon starvation requires specific enzymes for the depolymerization of polymers. In this sense, only one gene, *EGO55_09590*, codes a PHA depolymerase in *C. tardaugens* genome, sharing 45% sequence amino acid identity with PhaZ1 from *C. necator*.

Interestingly, previous transcriptomic experiments performed in *C. tardaugens* growing in pyruvate, with TES and E2 as carbon sources under no nitrogen limitation [6,9], found that the *EGO55_12100* gene, encoding the putative PhaP phasin, showed an induction of 34- and 18-folds on TES and E2 containing media, respectively, compared to cells growing on pyruvate (Appendix A), suggesting that this protein can play a fundamental role under TES and E2 growing conditions.

### 3.2. Production of PHAs from Steroids in C. tardaugens NBRC 16725

Based on in silico and transcriptomic analyses, we determined if *C. tardaugens* was capable of producing PHAs in minimum medium M63 0.1 N containing E2 or TES as carbon sources, compared to NB rich medium. When cells at early stationary phase were observed by TEM, we detected the presence of PHA granules (Figure 1), demonstrating that the strain produces PHAs in the conditions tested.

GC-MS analysis of the PHA extracted by methanolysis revealed the presence of a PHA copolymer formed by 3HB and 3HV (Figure 2). This analysis showed differences in quantity and composition in the three growing conditions studied. Maximum accumulation was achieved when cells were cultured in the presence of TES (47.2%), followed by cells growing in E2 (30.2%), and finally in rich medium NB (8.8%). Moreover, the PHAs produced using TES or E2 as carbon sources showed a higher proportion of PHV%/PHB%, i.e., 85.6/14.3 and 62.2/37.7, respectively, when compared to the PHA produced in NB that was 4.5/95.4 (Figure 2). The biomass reached using TES, E2 and NB as substrates was 0.41, 0.35 and 0.54 g L^−1^, respectively, and the PHA productivity after 48 h was 5.0, 2.0 and 0.8 mg L^−1^ h^−1^, respectively. Nevertheless, it is expected that this productivity can be further increased after process optimization on bioreactors.

The structure of the copolymer was further confirmed by nuclear magnetic resonance (NMR) (Figure 3). In the spectrum, for both PHBV copolymers formed in M63 0.1 N minimal medium supplemented with 2 mM E2 (Figure 3c) and formed in M63 0.1 N minimal medium supplemented with 1.89 mM TES (Figure 3d), the peaks of methyl protons (-CH_3_) of the PHB side group (C) appeared to have a doublet resonance at 1.2 ppm, methyl protons (-CH_3_) of PHV side group (G) at 0.83 ppm with triplet resonance, methylene protons (-CH_2_) of PHV side group (F) at 1.56 ppm with multiplet resonance, methylene protons (-CH_2_) of PHBV main chain (A and D) at 2.50 ppm, and -CH of PHV (E) and -CH of PHB (B) appeared to have a multiplet resonance at 5.1 ppm and 5.18 ppm, respectively. The results of ^1^H-NMR analysis showed that the copolymers were PHBV, which is in full agreement with the results of GC-MS analysis and with the ^1^H-NMR of the commercial standards. The chemical shift signals were in agreement with the literature [40,41]. Moreover, the COSY spectrum (Appendix A) corroborated the structure, showing the hydrogens that are coupled to each other.

### 3.3. Distribution of PHA Genes in Phylogenetically Related Bacteria

The search for proteins homologous to those previously identified in *C. tardaugens* was extended to include *Sphingopyxis alaskensis* LMG 18877 [27], *Sphingopyxis chilensis* S37 [27] and *Novosphingobium nitrogenifigens* Y88^T^ [42], bacteria belonging to the Sphingomonadaceae family described as PHA producers. Within this family, there are also other bacteria capable of synthesizing PHAs, such as *Novosphingobium* sp. THA_AIK7 [26]; however, its genome is not available in the databases. We have found homologous proteins encoded in the genomes of *S. alaskensis* LMG 18877 and *N. nitrogenifigens* Y88^T^ strains (Appendix A), suggesting that both strains should share a common PHA metabolism with *C. tardaugens*. It should be noted that the bacterium *S. chilensis* S37 does not contain any gene homologous to those included in the search.

Likewise, the search for proteins homologous to those previously identified in *C. tardaugens* was extended to other estrogen-degrading bacteria. Bacteria whose estrogen-degrading capacity has been described but that are not available as a search database in the BLASTp tool were excluded from the search. We have found homologous proteins encoded in the genomes of almost all of the species included in Appendix A, suggesting that these bacteria might potentially synthesize PHAs using estrogens as carbon source. It should be noted that *Vibrio* sp. H5, described as estrogen degrader bacterium, does not contain any gene homologous to those included in the search. Likewise, *Steroidobacter denitrificans* FST lacks a protein homologous to the PHB oligomer hydrolase PhaY. *S. denitrificans* FST and *Denitratisoma oestradiolicum* DSM 16959 lack PhaZ homologs and *Acheomobacter xylosoxidans* NBRC 15126, *Sdo. denitrificans* FST and *D. oestradiolicum* DSM 16959 lack PhaC homologs.

## 4. Discussion

PHAs play a key role in the physiology of natural producers, being essential for survival in the environment. Interest in PHAs focuses on their properties and production conditions as eco-friendly thermoplastics that can be obtained from renewable carbon resources. Bacteria produce PHAs as carbon and energy-storage materials that can be completely biodegraded to CO_2_ and H_2_O [15]. In addition, the ability of PHA accumulation by bacteria is of interest for various biotechnological applications, including bioremediation [22]. In this work, we explored PHA production from steroids that are found contaminating aquatic environments.

In this work, we demonstrated that *C. tardaugens* can accumulate PHA as intracellular storage granules while growing on E2 or TES as the sole carbon source. The amount of PHA produced was 47.2% PHA/dry weight in TES, 30.2% in E2, and only 8.8% in NB rich medium, suggesting that maximum production requires limiting nitrogen conditions as described for *C. necator* [43].

The analysis of *C. tardaugens* genome has provided a comprehensive view of PHA metabolism in this bacterium. The phasin and PHA synthase coding genes *EGO55_12100* and *EGO55_12105*, respectively, form a putative operon. Moreover, the acetoacetyl-CoA reductase and PHA synthase coding genes *EGO55_18540* and *EGO55_18530* are clustered together, separated by a putative TetR-family transcriptional regulator, encoded by *EGO55_18535*. However, the other genes involved in PHA biosynthesis in *C. tardaugens* are located along the chromosome and do not form a gene cluster, as occurs in *C. necator* [44]. *C. necator* is equipped with a high-efficiency *phaC_1_AB_1_R* operon, encoding the key enzymes PHA synthase (*phaC1*), acetyl-CoA acetyltransferase (*phaA*) and acetoacetyl-CoA reductase (*phaB1*), and a transcriptional regulator (*phaR*) [45]. According to the number of gene homologs found in *C. tardaugens* genome, we anticipate some significant differences between the metabolism of PHA in *C. tardaugens* and *C. necator*. Although the contribution of some gene products to PHB synthesis remains to be investigated, the genomic sequence of *C. necator* shows great redundancy in the genes involved in PHB synthesis, i.e., 37 *phaA*, 15 *phaB*, 2 *phaC*, 4 phasin encoding genes, 7 *phaZ* and 2 *phaY*. *C. tardaugens* genome contains a reduced number of these genes, i.e., 3 *phaC*, 9 *phaA*, 2 *phaB*, 1 phasin encoding gene, 1 *phaZ* and 1 *phaY*.

The highest expression levels observed in PHA biosynthesis genes of *C. tardaugens* when grown in steroids corresponded to the putative phasin PhaP encoded by *EGO55_12100*, which is homologous to the PhaP encoded by *Y88_3019* in the genome of the PHA-producing bacterium *N. nitrogenifigens* Y88^T^. Phasins are a class of low-molecular-mass amphipathic proteins that form a layer at the surface of the PHB granule to stabilize it, being the most abundant granule-associated proteins under PHA production conditions. This result is in accordance with the results reported by other authors, where the phasin encoded by *Y88_3019* gene was among the most abundant enzymes found in the proteome, under the studied conditions [42]. There is also a slight induction (3–5 fold) in some of the PHA biosynthesis genes (Appendix A), suggesting their involvement in PHA production in those conditions.

PHB has poor physical properties in terms of brittleness and low processability due to its highly crystalline nature [15,46], and because of that, the use of PHA copolymers has been considered as an alternative to improve these properties, with PHBV as the most useful one. An increase in the percentage of 3-hydroxyvalerate in PHBV decreases the degree of polymer crystallinity, reducing the melting temperature (Tm) and brittleness, and facilitates its industrial processing [47]). As it is known, the type of polymer produced depends on the carbon sources available, the flexibility of the organism’s intermediary metabolism, and the substrate specificity of the PHA biosynthetic enzymes [48]. In this sense, *C. tardaugens* is shown to produce a great amount of PHBV when the steroid catabolism provides the required building blocks. The production of 3HV moiety of PHBV requires the condensation of acetyl-CoA and propionyl-CoA to form 3-ketovaleryl-CoA (Figure 4). Biochemical analyses determined that BktB is the primary condensation enzyme leading to production of 3-hydroxyvalerate derived from propionyl-CoA in *C. necator*. *EGO55_09255* coding gene shares 60% amino acid identity with BktB, and it is induced 3.6- and 2.3-folds under TES and E2 growing conditions, respectively, compared with pyruvate growing conditions [6,9] (Appendix A), suggesting that this gene could be involved in the production of 3-ketovaleryl-CoA. Steroid catabolism renders acetyl-CoA, succinyl-CoA and propionyl-CoA. Acetyl-CoA and succinyl-CoA enter central metabolism through the TCA cycle, and propionyl-CoA has to be metabolized either by the methylcitrate cycle or by the methylmalonyl-CoA pathway. The genome analysis of *C. tardaugens* revealed that it does not possess the methylcitrate cycle, and, therefore, it metabolizes propionyl-CoA through the methylmalonyl-CoA pathway [6]. In fact, the transcriptomic analysis showed high levels of induction in the genes involved in the methylmalonyl-CoA pathway on TES and E2, suggesting the existence of an excess of propionyl-CoA under these growing conditions [6,9].

Figure 2 shows that *C. tardaugens* produces PHBV in steroid growing conditions much more efficiently than when growing in rich medium. This was expected, since PHB accumulation is optimal under an excess of carbon and limiting levels of nitrogen, phosphorous or oxygen [49,50].

The search for PHA biosynthesis genes homologous to that from *C. tardaugens* in other estrogen-degrading bacteria showed that the putative PHA genes are widely distributed. As some of those bacteria have an estrogen degradation gene cluster homologous to that involved in E2 degradation in *C. tardaugens* [6], both steroid and PHA metabolism might play a key role during survival of these strains in the niches where they were isolated.

Regarding the localization of the PHA genes in the genome, *EGO55_12100* and *EGO55_12105*, encoding PhaP and PhaC, are clustered together in *C. tardaugens*. Interestingly, that distribution is also shared by their homologous genes in *Croceicoccus estronivorus*, *Sphingobium bisphenolivorans* YL23, *S. alaskensis* LMG 18877 and *N. nitrogenifigens* Y88. The other genes involved in PHA metabolism are distributed throughout the genome in these bacteria.

Finally, the finding that steroid degrader bacteria can be used not only to eliminate endocrine disruptor contaminants, but also to produce PHAs, and particularly PHBV copolymers, using steroids as carbon sources, opens new avenues to investigate the use of abundant natural sterols such as cholesterol or bile acids as putative substrates for the production of bioplastics.

## Figures and Tables

**Figure 1 microorganisms-10-00706-f001:**
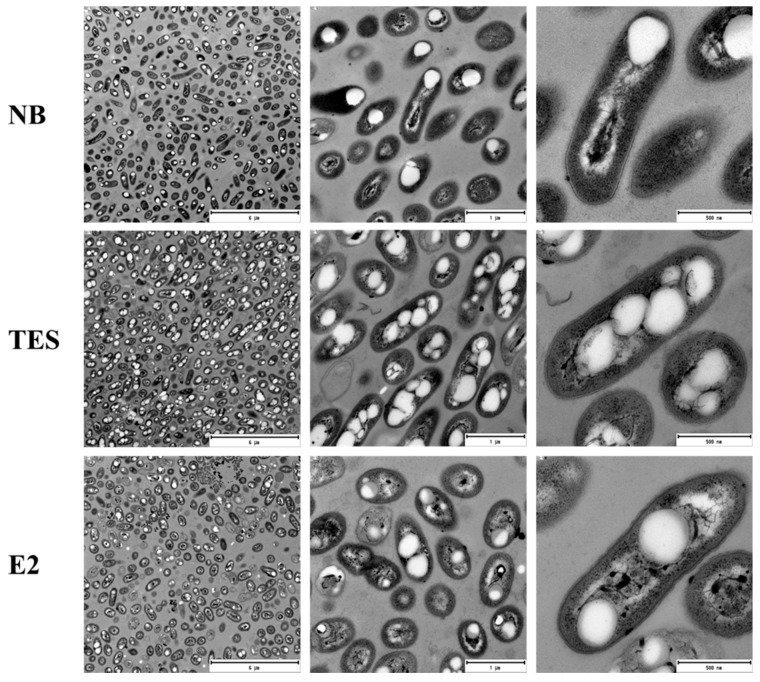
Transmission electron microscopy (TEM) images of *C. tardaugens* NBRC 16725. PHAs are observed as granules in the cells growing in NB rich medium and in M63 0.1 N minimal medium supplemented with 1.89 mM TES or 2 mM E2. From left to right the scale bars are 6 µm, 1 µm and 500 nm for each condition.

**Figure 2 microorganisms-10-00706-f002:**
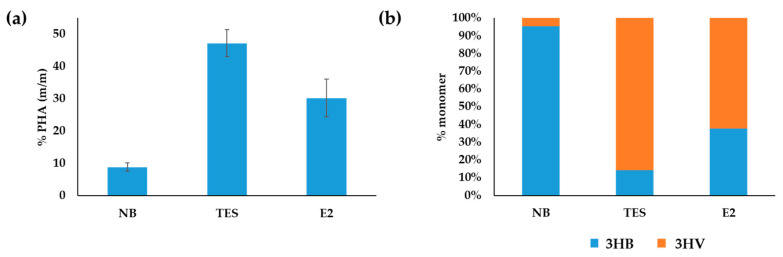
GC-MS analysis of PHA production in *C. tardaugens* NBRC 16725. (**a**) Accumulated PHA (*m*/*m*) as % of cell dry weight. (**b**) Proportion of PHB and PHV in the PHA copolymer. The data correspond to the mean of three independent biological replicates.

**Figure 3 microorganisms-10-00706-f003:**
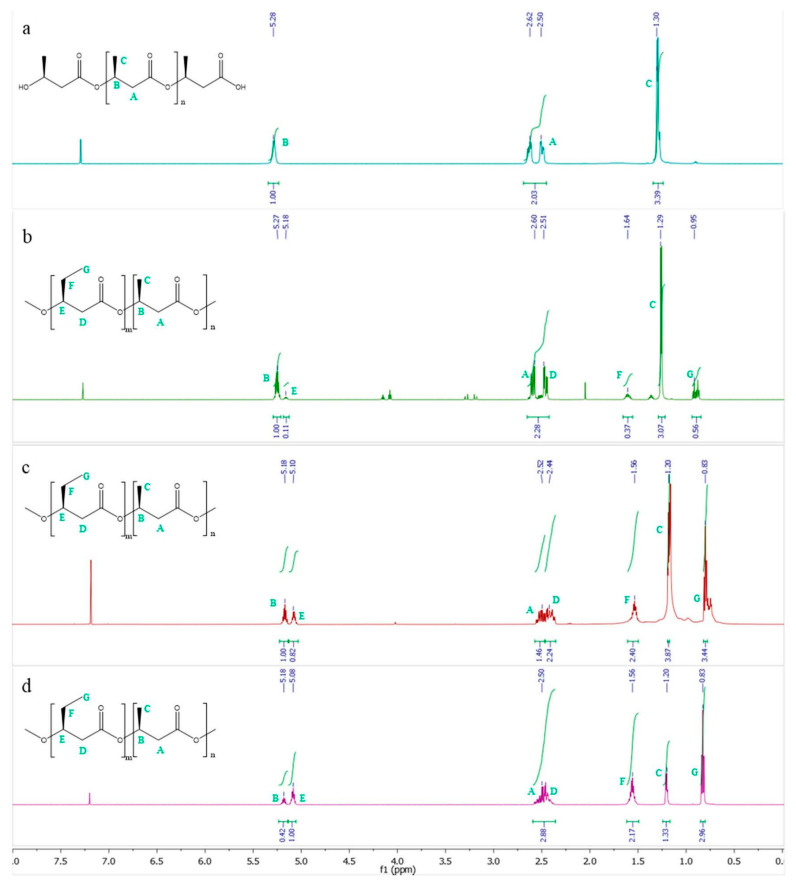
NMR analysis of the PHBV copolymer produced by *C. tardaugens* NBRC 16725. (**a**) PHB commercial standard, (**b**) PHBV commercial standard, (**c**) PHBV copolymer formed in M63 0.1 N minimal medium supplemented with 2 mM E2. (**d**) PHA copolymer formed in M63 0.1 N minimal medium supplemented with 1.89 mM TES.

**Figure 4 microorganisms-10-00706-f004:**
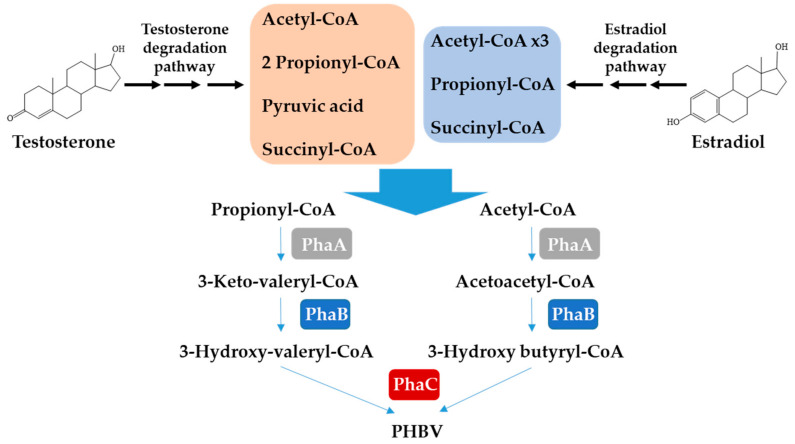
Metabolic pathways involved in production of PHBV copolymer from steroidal endocrine disruptors in *C. tardaugens*. The central metabolites derived from the catabolism of TES and E2 are shown in the brown and blue boxes, respectively. PhaA, β-ketothiolase. PhaB, 3-ketoacetyl-CoA reductase. PhaC, PHA synthase.

**Table 1 microorganisms-10-00706-t001:** In silico identification of genes involved in PHA metabolism in *C. tardaugens* NBRC 16725 (accession number CP034179).

Gene Name	Function	Gene Locus (*EGO55_*)
PhaA	β-ketothiolase	*09255*
		*20150*
		*05760*
		*05015*
		*13750*
		*13790*
		*05665*
		*03475*
		*14320*
PhaB	acetoacetyl-CoA reductase	*18540*
		*09490*
PhaC	PHB synthase	*12105*
		*17580*
		*18530*
PhaZ	PHB depolymerase	*09590*
PhaP	Phasin	*12100*

## Data Availability

Not applicable.

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
