# Peer review of "Polyhydroxyalkanoate Production by Caenibius tardaugens from Steroidal Endocrine Disruptors"

_microorganisms, 2022, doi:10.3390/microorganisms10040706_

Round 1
Reviewer 1 Report
In this work, the authors investigate the production of polyhydroxyalkanoate by Caenibius tardaugens from steroidal endocrine disruptors. This manuscript needs major revision before publication.
Introduction. There are several missing references in the PHA production paragraph. For instance: Lines 42-45, “These materials have been actively studied as promising materials for solving issues related to the currently used plastics because they possess physical and thermochemical material properties comparable to those of petroleum-based plastics.”; Lines 51-52, “There is an increasing interest to produce these polymers using 51 different wastes as substrates mainly with the aim of reducing PHA production costs”. Please add some recent review articles on PHA description, production and application.
Line 72, “(Godoy et al., 2003a; Addison et al., 2007; Teeka et al., 2012),” please number these references.
Materials and Methods.
Strains and culture conditions
Please add more details about the culture conditions of C. tardaugens such as total volume of the culture, type of reactor, and time.
Results
Authors should present results on C. tardaugens growth and PHA productivity under the tested conditions.
Line 212. “Figure 2 (a) Accumulated 212 PHA (m/m)” m/m? Is this % of cell dry weight.
Discussion
I suggest that the authors need to expand the discussion section.
Please add the metabolic pathways (schematic) for the use of steroidal endocrine disruptors for PHB biosynthesis by C. tardaugens.
There are several missing references in this section. For example: Line 260, “Bacteria produce PHAs as carbon and energy-storage materials”; Line 262, “various biotechnological applications including bioremediation”; Line 302, “and the substrate specificity of the PHA biosynthetic enzymes”
Line 323. Replace “degrade” with “degrading”
Author Response
Reviewer 1
In this work, the authors investigate the production of polyhydroxyalkanoate by Caenibius tardaugens from steroidal endocrine disruptors. This manuscript needs major revision before publication.
Introduction. There are several missing references in the PHA production paragraph. For instance: Lines 42-45, “These materials have been actively studied as promising materials for solving issues related to the currently used plastics because they possess physical and thermochemical material properties comparable to those of petroleum-based plastics.”; Lines 51-52, “There is an increasing interest to produce these polymers using 51 different wastes as substrates mainly with the aim of reducing PHA production costs”. Please add some recent review articles on PHA description, production and application.
Line 72, “(Godoy et al., 2003a; Addison et al., 2007; Teeka et al., 2012),” please number these references.
All the requests have been addressed in the new version
Materials and Methods.
Strains and culture conditions
Please add more details about the culture conditions of C. tardaugens such as total volume of the culture, type of reactor, and time. It has been added as requested
Results
Authors should present results on C. tardaugens growth and PHA productivity under the tested conditions. A paragraph has been added in the result section to clarify this point.
Line 212. Figure 2 (a) Accumulated PHA (m/m)” m/m? Is this % of cell dry weight. We have explained it in the figure legend.
Discussion
I suggest that the authors need to expand the discussion section. Some aspects of the discussion section have been expanded as requested.
Please add the metabolic pathways (schematic) for the use of steroidal endocrine disruptors for PHB biosynthesis by C. tardaugens. We have included figure 4 showing a schematic representation of the main aspects of the metabolic pathways involved in PHA production from steroidal compounds.
There are several missing references in this section. For example: Line 260, “Bacteria produce PHAs as carbon and energy-storage materials”; Line 262, “various biotechnological applications including bioremediation”; Line 302, “and the substrate specificity of the PHA biosynthetic enzymes” The required references have been added.
Line 323. Replace “degrade” with “degrading” Done as requested
Reviewer 2 Report
Recommendation: Publish after minor revisions noted.
Juan et al described the α-proteobacterium Caenibius tardaugens can use estrogens and androgens as the sole 7 carbon source to produce polyhydroxyal-10 kanoates (PHA). The manuscript is clear and interesting. I recommend publication with minor revisions.
Add the 1H NMR data for PHBV, PHA copolymer formed and commercial standard.
Author Response
Reviewer 2
Recommendation: Publish after minor revisions noted.
Juan et al described the α-proteobacterium Caenibius tardaugens can use estrogens and androgens as the sole carbon source to produce polyhydroxyalkanoates (PHA). The manuscript is clear and interesting. I recommend publication with minor revisions.
Add the 1H NMR data for PHBV, PHA copolymer formed and commercial standard. This information has been included in Figure 3.
Round 2
Reviewer 1 Report
The authors performed the requested changes and corrections. The paper is now suitable for publication.